# Serum Folate Concentrations in Exclusively Breastfed Preterm Infants Who Received No Supplementary Oral Folic Acid After Discharge: A Prospective Cohort Study

**DOI:** 10.3390/nu16234220

**Published:** 2024-12-06

**Authors:** Isabel Iglesias-Platas, Agata Sobczyńska-Malefora, Vennila Ponnusamy, Ajit Mahaveer, Kieran Voong, Amy Nichols, Karen Dockery, Nicky Holland, Shaveta Mulla, Martin J. Shearer, David Card, Lindsay J. Hall, Dominic J. Harrington, Paul Clarke

**Affiliations:** 1Neonatal Intensive Care Unit, Norfolk and Norwich University Hospitals NHS Foundation Trust, Norwich NR4 7UY, UK; amy.nichols@nnuh.nhs.uk (A.N.); shaveta.mulla1@nhs.net (S.M.); 2Nutristasis Unit, Synnovis, Guy’s and St. Thomas’ Hospital NHS Trust, London SE1 7EH, UK; agata.malefora@synnovis.co.uk (A.S.-M.); kieran.voong@synnovis.co.uk (K.V.); david.card@nhs.net (D.C.); dominic.harrington@synnovis.co.uk (D.J.H.); 3Faculty of Life Sciences and Medicine, King’s College London, London WC2R 2LS, UK; 4Neonatal Intensive Care Unit, Ashford and St Peter’s Hospital, Chertsey KT16 0PZ, UK; vennilaponnusamy@nhs.net (V.P.); nicky.holland1@nhs.net (N.H.); 5Neonatal Intensive Care Unit, Manchester University NHS Foundation Trust, Manchester M13 9WL, UK; ajit.mahaveer@mft.nhs.uk (A.M.); karen.dockery@mft.nhs.uk (K.D.); 6Centre for Haemostasis and Thrombosis, Guy’s and St. Thomas’ Hospital NHS Trust, London SE1 7EH, UK; martin.shearer@gstt.nhs.uk; 7Institute of Microbiology and Infection, College of Medical and Dental Sciences, University of Birmingham, Birmingham B15 2TT, UK; l.hall.3@bham.ac.uk; 8Gut Microbes and Health, Quadram Institute Bioscience, Norwich Research Park, Norwich NR4 7UG, UK; 9Norwich Medical School, University of East Anglia, Norwich NR4 7TJ, UK; 10School of Biosciences and Medicine, University of Surrey, Guilford GU2 7XH, UK

**Keywords:** preterm infant, folic acid supplementation, breastfeeding, breastmilk fortifier, parenteral nutrition

## Abstract

**Background/Objectives:** Adequate folate intake is required in preterm infants for rapid growth and development, but there is little evidence to back recommendations. We aimed to assess folate status in preterm infants at discharge and in early infancy, according to exposure to folate sources, particularly in those exclusively/predominantly breastfed. **Methods:** A prospective, multicenter, observational cohort study was conducted in the UK, involving 45 preterm infants <33 weeks’ gestational age (GA) exclusively/predominantly fed human milk when approaching NICU (Neonatal Intensive Care Unit) discharge. Serum folate levels were measured near NICU discharge (T1) and at 2–3 months corrected age (T2). Folate status was categorized per WHO (World Health Organization) guidelines: deficiency (<6.8 nmol/L), possible deficiency (6.8–13.4 nmol/L), normal (13.5–45.3 nmol/L), and elevated (>45.3 nmol/L). Nutritional information on feed and supplements was collected from hospital notes and maternal interviews. **Results:** Thirty-two infants (71%) received parenteral nutrition. Twelve infants (32%) remained exclusively breastfed at T2. No infant from the whole cohort had a serum folate concentration <13.5 nmol/L at either time point. A proportion of infants had serum folate concentrations >45.3 nmol/L: 14/45 (31%) at T1, 19/37 (42%) at T2, and 7/37 (16%) at both time points. Elevated concentrations were seen particularly in infants who received folic acid supplements or nutrition containing folic acid, such as parenteral nutrition and breastmilk fortifiers. **Conclusions:** Folate deficiency was not observed in this cohort; folate concentrations were high and in line with those observed in healthy term infants. Further research is needed to assess the high folate concentrations in premature babies and whether they may have any adverse clinical impact.

## 1. Introduction

The prenatal period exhibits the highest rate of bodily growth across the lifespan of any individual. In the case of preterm birth, a fraction of this stage takes place in the neonatal unit, and it partially depends on adequate nutritional support [1]. Recommendations for macronutrient supply in this population are based on a solid body of evidence, whereas guidelines for vitamin supply [2,3] are mostly agreed upon by expert consensus, due to the lack of well-designed studies. Current commercial products aimed at provision of parenteral nutrition (PN) or milk fortification addressing this group of newborns are designed to comply with these guidelines. However, preterm babies being exclusively fed their mothers’ own milk (MOM) during admission or after discharge home often receive marketed multivitamin products that are not specifically tailored for this population and which can therefore lack some essential micronutrients [4]. This might entail a risk of subclinical or even symptomatic deficiency before discharge or during early infancy, as recently shown for vitamin K [4,5].

Infancy may be associated with a risk of folate depletion due to the high demands of rapid growth. Preterm infants may be particularly susceptible because of the demands of rapid red blood cell production and multifactorial anemia of prematurity. Folate is essential for both processes. Previous research has shown that preterm infants are at risk of developing folate deficiency, if defined based on concentrations below the threshold considered normal for adults [6,7].

Folate is one of the vitamins in the B group and is also known as vitamin B9. Multiple natural folate forms exist which are required for major cellular processes including methylation reactions, amino acid metabolism, and nucleic acid synthesis. Folic acid is a synthetic compound and is most commonly used in supplements and fortified foods. The folate cycle (one carbon metabolism) is closely intertwined with the vitamin B12 pathway, and a deficiency in B12 may preclude folate (5-methyltetrahydrofolate form) from being converted to tetrahydrofolate, a form of folate essential for nucleic acid synthesis. Therefore, both folate and vitamin B12 are particularly relevant in periods of rapid growth or in tissues with high cellular turnover, hence their essential role in development. It has been known since the 1990s [8] that folate deficiency is associated with neural tube defects. This has been the basis for the mandatory food fortification policies of many countries (not currently including the United Kingdom) and for offering routine folic acid supplementation to women planning pregnancy.

Serum folate concentrations are usually high at birth in healthy term infants, surpassing maternal levels, and subsequently decline with time [9]. The reason for this postnatal physiological decline is not known but could be related to the expansion of blood volume and to the storage of folate in tissues. A few landmark studies have shown that this is mirrored in preterm infants as well, and that, without supplementation, some babies will have serum/plasma folate concentrations below the lower threshold of what is considered normal in adults [6,7,10]. The European Society for Paediatric Gastroenterology, Hepatology and Nutrition (ESPGHAN) currently recommends an enteral intake of 23–100 µg/kg/day of folate in preterm infants [3]. This range is based on studies demonstrating improvement in clinical outcomes with supplementation or intakes from infants who are asymptomatic, do not suffer health deterioration, or show correction of a nutrient biomarker in their blood [3]. Despite high folate bioavailability due to the presence of folate-binding proteins [11], folate concentration in human milk is low [12,13]. Based on this, together with the observation that preterm infants given a supplement of 50 µg/day, and no other nutritional source of folate, had folate concentrations comparable to breastfed term infants [14], local and regional guidelines (e.g., East of England vitamin guideline [15]) suggest, since 2018, the provision of a further 50 µg of folic acid/day until the expected due date to preterm babies fed exclusively unfortified breastmilk [15]. Infants receiving commercial breastmilk fortifiers or preterm formulas receive folic acid intakes well within recommended ranges because these contain supplementary folic acid. A previous contemporary study suggested that extra folic acid supplementation during admission in the neonatal unit might not be necessary for very preterm infants up until about 36 weeks’ postmenstrual age independent of the mode of feeding [16]. This premise potentially reflects the replenishment of folate stores by the use of initial nutritional products and an increased stability of folate contained in milk [17].

However, the question remains as to whether folate deficiency might develop later in time in breastfed babies when exposure to folic acid-supplemented nutritional products (PN, fortifiers) given in the neonatal intensive care unit (NICU) has ceased.

We sought to investigate serum folate concentrations in preterm infants at the time of discharge and post discharge. Our study predated the recommendations to give routine extra oral folic acid supplementation during the NICU stay; we focused particularly on those being fed unfortified breastmilk exclusively.

## 2. Materials and Methods

This was a prospective, multicenter, observational UK study to assess the micronutrient status of exclusively breastfed preterm infants approaching the time of discharge home and in early infancy. This study had research ethics approval (REC Ref. No. 15/LO/1808) and parents provided written informed consent prior to entry. The primary outcome was the prevalence of functional vitamin K insufficiency and those results are published elsewhere [4]. The present study reports the secondary outcome of folate status in the same cohort.

In brief, families of preterm infants born before 33 weeks of gestation in four UK NICUs were approached for participation near the time of NICU discharge if the babies were being fed human breastmilk exclusively or predominantly (arbitrarily defined as >80% of overall volume intake being human breastmilk) and mothers intended to continue/achieve full breastmilk feeding at home. Infants with cholestatic jaundice were excluded. Participants were recruited between January 2016 and April 2018.

Nutritional practices were in line with the international guidance at that time: PN was started as soon as possible (routinely in infants < 30 weeks’ gestation or weighing < 1250 g at birth) as was enteral nutrition, with the incremental progression of milk volumes and weaning of PN until full enteral feeds were established (defined as milk intake of ≥150 mL/kg/day). As part of PN therapy, all infants received a vitamin supplement (Solivito N^®^, Fresenius Kabi Ltd., Runcorn, UK), which provided 10–40 µg/kg/day of folic acid, depending on the rates of infusion. For exclusively breastfed babies, the routine practice at all participating centers was routine fortification of their MOM using a commercial breastmilk fortifier (BMF) powder (Cow & Gate Nutriprem Human Milk Fortifier, Nutricia Ltd., Trowbridge, UK) once full enteral feeds were reached; fortifier provided an extra 30 µg of folic acid in the amount recommended for each 100 mL of human milk (which would amount to 45–60 µg/kg/day of folic acid at the average daily feed volumes taken in of 150 mL/kg/day). Fortification was discontinued before discharge. Artificial preterm milk formula (Nutriprem 1, Cow & Gate, Nutricia Ltd., Trowbridge, UK) contains ~60 µg folate/100 mL. Since 2018, and following international recommendations, UK guidelines recommended providing 50 µg/day of oral folic acid solution to exclusively breastfed preterm infants when an enteral feed volume of 100 mL/kg/day was reached and <5 mL/kg/day of intravenous lipid was being given, up until the baby’s expected due date. At the time of this study, this recommendation had been implemented in only one of the four participating NICUs, while the other centers did not provide extra folic acid supplementation before or after discharge during the recruitment period.

*Study procedures*: Blood samples were obtained from participants at two time points: T1, pre-discharge while still in-patient in the NICU, and T2, on follow-up in early infancy during an outpatient visit at approximately 2–3 months’ corrected age (CA). Nutritional information on the type of feed and supplements was collected from hospital notes and via maternal interviews. The time of sampling was not standardized with respect to the timing of feeds.

*Sample processing*: Blood samples were sent to the local hospital biochemistry laboratory for separation. Serum samples were stored in a −80 °C freezer and shipped in batches on dry ice to a facility performing folate measurements (Nutristasis Unit, London, UK).

*Laboratory measurements*: Serum folate was determined by a chemiluminescent microparticle folate binding protein assay, with a reading range of 7.0–45.3 nmol/L (Abbott Diagnostics, Chicago, Illinois, United States of America). World Health Organization (WHO) guidelines for the general population were used to define deficiency (<6.8 nmol/L), possible deficiency (6.8–13.4 nmol/L), normal (13.5–45.3 nmol/L), and elevated folate status (>45.3 nmol/L) [18] due to the lack of specific reference ranges for infants.

*Statistical analysis*: For the purposes of analysis, serum folate concentrations that exceeded the linear range of the assay (45.3 nmol/L) were assigned a value of 45.5 nmol/L. Appropriate statistical tests were used as applicable to evaluate and describe differences between groups: Student’s T/ANOVA for continuous variables; χ^2^ or Fisher’s exact test for categorical variables; and Mann–Whitney U/Kruskal–Wallis tests for continuous variables with non-normal distribution. Bonferroni correction was used for post hoc comparisons between multiple groups as appropriate. Repeated measures tests were applied when comparing the two time point measurements for the same patient. Statistical significance was set at a 2-sided *p*-value < 0.05. Variables significantly related to serum folate levels in univariate analysis (with *p*-values < 0.1) were included in the multivariate analysis to account for the role of confounding factors; this was undertaken by stepwise backwards linear regression. All analyses were performed using IBM SPSS Statistics software version 29.0 (IBM Corp., Armonk, NY, USA).

## 3. Results

### 3.1. Baseline Characteristics, Feeding, and Folic Acid Supplements

A total of 45 infants were recruited during the study period. A first blood sample (T1) was taken from 45 infants at a median postmenstrual age of 35.1 (IQR 34.9–36.4) weeks, and 37 infants returned for the second visit (T2) at a median CA of 9.3 (IQR 4.7–16.0) weeks. Thirty-two infants (71%) received PN for a median duration of 10.0 (IQR 7.0–14.8) days. There was a high attrition rate for exclusive breastmilk feeding post discharge and only 12 infants (32%) remained exclusively breastfed at T2. Feeding trajectories are represented in Appendix A. Baseline characteristics and nutritional support are summarized in Table 1 and specific information per each feeding group can be found in Appendix A.

### 3.2. Serum Folate Results

Median (interquartile range; IQR) serum folate concentrations were 39.2 (37.4–45.5) nmol/L at T1 and 43.0 (41.7–45.5) nmol/L at T2 (Figure 1). No infant had a serum folate concentration < 13.5 nmol/L at either time point. A proportion of infants had serum folate concentrations > 45.3 nmol/L: 14/45 (31%) at T1, 19/37 (42%) at T2, and 7/37 (16%) at both time points.

There were no differences in T1 serum folate concentrations between infants who received PN and those who did not (PN, n = 32, median serum folate: 43.5 nmol/L, IQR 37.8–45.5 nmol/L vs. no PN, n = 13, median serum folate: 41.0 nmol/L, IQR 25.6–44.6 nmol/L, *p* = 0.15), although this might have been due to small numbers as well as not having had more accurate serum folate estimations for specimens with values >45.3 nmol/L, as there was a positive correlation between days on PN and serum folate concentrations at T1 (Spearman’s rho: 0.303, *p* = 0.04). We did not see any correlation between serum folate levels at T1 and weight at that time point (Spearman´s rho: –0.017, *p* = 0.913) or Z-score for weight at T1 (Spearman´s rho: 0.101, *p* = 0.500). Serum folate concentration at T2 was not correlated with weight at T2 either (Spearman´s rho: 0.274, *p* = 0.100). Enteral feeding was associated with significant differences in serum folate concentrations at both T1 and T2 (Figure 1, Panel 3 and Table 2). At T1, the highest concentrations were found in the group that received fortified breastmilk (median 44.4 nmol/L, IQR 41.7–45.5 nmol/L) as compared to mixed feeding infants (median 36.0 mmol/L, IQR 29.9–43.0 nmol/L) or unfortified MOM infants (median 19.5 nmol/L, IQR 15.9–32.8 nmol/L) (*p* < 0.001). The group of six infants with the lowest serum folate concentrations (who are indicated in Figure 1, panels 1 and 3) were of higher gestational age (median 31.7 weeks, IQR 30.2–32.3 weeks vs. 29.7 weeks, IQR 26.1–31.0 weeks, *p* = 0.024) and of lower chronological age at T1 (median 22.5 days, IQR 15.0–33.3 days vs. median 45.0 days, IQR 27.8–63.5 days). They received fewer days of parenteral nutrition (medians: 3, IQR 0–7 days vs. 8, IQR 0–13 days) and comprised a higher proportion of individuals that did not receive breastmilk fortifier prior to T1, 6/6 (100%) vs. 4/39 (10.3%) in the rest of the group. At T2, folate concentrations were highest in those babies on cows’ milk-based artificial formula (median 45.3, IQR: 45.5–45.5 nmol/L), followed by those on mixed feeding (median 45.5, IQR: 43.9–45.5 nmol/L), then those exclusively breastfed (median 39.0 nmol/L, IQR 34.0–43.5 nmol/L) (*p* < 0.001), although the difference was not significant between mixed and artificial feeding in a post hoc analysis (MOM vs. Mixed: adjusted *p* = 0.01; MOM vs. Formula: adjusted *p* < 0.001; Mixed vs. Formula: adjusted *p* = 1.0). Folate levels >45.3 nmol/L were more frequent at T2 in formula-fed babies (Mother´s Own Milk: 1/12, 8%; Mixed feeding: 5/9, 56%; Formula feeding: 13/16, 81%, *p* < 0.001). All five babies who were given folic acid oral supplements had serum folate concentrations >45.3 nmol/L at both time points.

In a multivariate regression analysis, the number of days on BMF (standardized β = 0.57, *p* < 0.001) and having received folic acid drops (standardized β = 0.37, *p* = 0.004) were independently related to serum folate levels at T1 when adjusted for gestational age (GA) at birth, PN reception, and type of milk feeding (model summary: adjusted R^2^: 0.35, *p* < 0.001). Folate levels at T2 were related to the type of milk feeding at the second visit (standardized β = 0.66, *p* < 0.001) when adjusted by GA, number of days on BMF during admission, and oral folic acid supplementation (model summary: adjusted R^2^: 0.41, *p* < 0.001).

## 4. Discussion

In this cohort study, which recruited predominantly/exclusively breastfed preterm infants, predating the implementation of current recommendations for additional oral folate supplementation such that most infants therefore did not receive extra folic acid supplements, no infant had a serum folate concentration indicative of deficiency (<6.8 nmol/L) either pre-discharge or on follow-up at 2–3 months’ corrected age. Our findings therefore seem to contradict current guidelines that suggest extra folic acid supplements are required in preterm infants fed unfortified breastmilk.

Several factors will impact the baseline availability of folate to a baby born prematurely. On the one hand, local guidelines for supplementation from the preconception period to 12 weeks of pregnancy were implemented in the 1990s in the UK, with the aim of preventing neural tube defects [8]. On the other hand, over 80 countries worldwide have adopted a population approach, with national programs for grain fortification [19]. While the UK has not yet implemented mandatory folic acid supplementation (folic acid fortification of flour will become mandatory as of 1 October 2026 in the UK), folic acid-fortified foods, such as breakfast cereals, are nevertheless widely available [19]. These differences in approaches need to be considered when assessing both the applicability of guidelines and the generalizability of results from previous studies.

Commercially manufactured nutritional products aimed at preterm babies contain sufficient folic acid to ensure adequate intakes, as per current nutritional guidelines [3]. In line with a previous study conducted in Turkey [16], our results suggest that preterm infants who have received parenteral nutrition and/or breastmilk fortifier have adequate folate concentrations and are unlikely to become deficient even if subsequently fed unfortified breastmilk. Our report has the added value of post discharge follow-up, demonstrating that deficiency did not develop even after an extended period of exclusive breastfeeding. However, both cohorts comprised too few cases that had not been exposed to any nutritional supplementation to draw firm conclusions regarding this subgroup, and further investigation may be needed in larger contemporary cohorts. We are aware that, as is the case in most nutritional studies, there will have been underlying differences between the groups with different types of feeding, as there are known factors that will influence the chances of successful breastfeeding. Random group distribution, however, was not possible for the obvious reason that it would not have been ethical to allocate infants to breastmilk vs. artificial formula feeding.

A sizeable proportion of preterm infants had elevated serum folate levels according to the WHO definition. Folate levels in infants are higher than in childhood and adulthood [20], and values in our cohort tended to be higher than those reported in healthy breastfeeding term infants at around 3 weeks and 4 months of age [20]. It is unclear whether preterm infants might require higher levels of circulating folate, but a Swedish study [21] suggested that estimated folate provision in parenteral and enteral nutrition correlated with growth outcomes in extremely preterm infants (<27 weeks’ GA), although there was no measurement of serum folate levels involved. We found no relationship between serum folate concentration at discharge and growth during admission, which is similar to what had been previously found in healthy breastfed term babies between 20 days and 4 months of age [20], but further studies are needed to explore the relationships between folate intake, biochemical measurements, and growth outcomes in these infants.

We found that serum folate levels increased between time points T1 and T2, even in exclusively breastfed infants who were not receiving folate from any other source. A rise in plasma folate from 3 to 6 months of age was also seen in a Norwegian cohort of term babies, where folate status was lower in breastfed compared to non-breastfed infants, yet none developed biochemical folate deficiency [22]. It is unlikely that folate levels in breastmilk in our cohort increased between time points, as folate concentrations in human milk have previously been shown to be stable and well preserved, except in cases of severe maternal deficiency [23]. Different blood levels with similar intakes have also been found in adults [24], suggesting that other factors were at play. In our case, this might have involved a variable abundance of folate-binding proteins in the breastmilk, which could have had an impact on bioavailability [25], or intestinal colonization by specific strains of folate-producing bifidobacteria in the infant [26].

There are a few limitations to our study. This study was carried out in a setting where folic acid food fortification had not yet been implemented, and this could be perceived as having had an impact on the generalizability of the results. We were unable to obtain complete information on maternal antenatal folic acid supplementation (with only a 50% response rate, though 100% of responders received folic acid supplements) and did not measure maternal serum folate levels. Nor did we measure the folate content of the maternal breastmilk. However, infant serum folate concentrations do not seem to correlate with maternal blood concentrations [20] and human milk folate concentrations seem not to be impacted by low-dose folate supplements [27]. In preterm infants, the impacts of maternal folic acid supplementation and other maternal characteristics seem to be limited to the early postnatal weeks [16], which were not assessed in our study.

We measured serum folate concentrations rather than red blood cell folate, which is considered to be a better reflection of folate stores. The British Committee for Standards in Haematology has stated that, at least in adults, serum folate concentration offers equivalent diagnostic capability for the assessment of folate status [28].

Due to ethical limitations on blood sampling volumes, as appropriate for these small infants, and corresponding low working serum sample volumes, we were unable to perform dilutions to explore just how high the serum folate concentrations were within the elevated folate category. Similarly, it was not possible to assess other metabolites related to the B12 and folate pathways (such as homocysteine, cobalamin, holo-transcobalamin, and methylmalonic acid), which would have facilitated a more robust interpretation of the biochemical impact of increased folate concentrations. Folate repletion has resulted in the uncovering of roles for B12 metabolism in other clinical situations [29], highlighting the tight interconnection between these two vitamins. Along the same lines, it would have been interesting to measure unmetabolized folic acid, which has been proposed as a biomarker [30] in the context of concerns about the potential adverse effects of excessive folic acid intake especially in conjunction with low vitamin B12 status [31]. This is particularly relevant in infancy, as B12 deficiency seems much more frequent in mothers and infants than folate deficiency, at least in term infants [22,32].

## 5. Conclusions

In conclusion, in a cohort of exclusively/predominantly breastfed preterm infants born <33 weeks of gestational age, and who were largely folate-unsupplemented post discharge, and using serum folate as a marker of status, we did not identify any cases of folate deficiency either before discharge from the NICU or on follow-up 2–3 months later. On the contrary, high serum folate concentrations, similar to those observed in healthy breastfed term infants, were frequent in infants who received supplementation in the form of folic acid-containing nutritional products (parenteral nutrition, breastmilk fortifiers, and formula) and were universal in the small proportion who received specific folic acid supplements. While this might indicate that current guidelines for folate intake in preterm infants can result in excessive levels, more information is needed to ascertain how this relates to other markers of folate/vitamin B12 pathways and whether it could relate to clinical outcomes.

## Figures and Tables

**Figure 1 nutrients-16-04220-f001:**
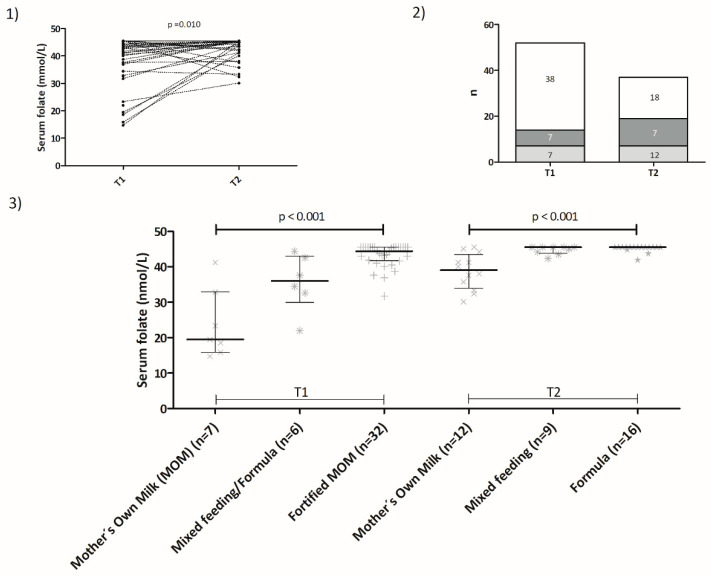
Folate status in preterm infants. (**1**) Serum folate concentrations for each participant at T1 (pre-discharge) and at T2 (follow-up). The dotted lines link the folate concentrations for each patient at each time point. Concentrations were significantly higher at T2 (n = 37 pairs, T1 mean 38.9, SD 9.1 nmol/L; T2 mean 43.0, SD 4.2 nmol/L, *p* = 0.01, repeated measures *t*-test). (**2**) Percentage and number of participants with folate concentrations that were normal, elevated at one time point, or elevated at both time points are represented by the white, light gray, and dark gray areas on the bars, respectively. (**3**) Serum folate concentrations according to feeding type at T1 and T2. Each individual symbol (x, ✴, +, ★) represents one data point for each feeding category. Lines and error bars represent medians and interquartile ranges, respectively. Mother’s Own Milk (MOM).

**Table 1 nutrients-16-04220-t001:** Baseline characteristics and nutritional support in the cohort.

Baseline Characteristics	
Gestational age at birth (weeks)	29.1 (2.8)//30.0 (26.4–31.4)
Birthweight (g)	1255 (433)//1337 (839–1588)
Gender: Male	20/45 (44%)
Fetal growth restriction	4/45 (9%)
Multiple (twin or triplet)	15/45 (33%) *
Postnatal age at blood sampling	
T1 (days)	46 (26)//41 (15–110)
T2 (days)	150 (52)//150 (69–238)
Nutritional data	
Parenteral nutrition	
Received	32/45 (71%)
Days **	11.2 (5.6)//10.0 (7.0–14.8)
Enteral feeding	
Days at start	1.9 (1.7)//2.0 (1.0–2.0)
Days at full feeds	11.5 (7.2)//10.0 (6.0–15.0)
Oral folic acid supplement	
Yes	5/45 (11.1%)
Total days where given	93 (29)//78 (72–100)
Type of feeding	
T1	
Unfortified breastmilk	7 (16%)
Fortified breastmilk	32 (71%)
Mixed feeding ***	6 (13%)
T2	
Breastmilk only	12 (32%)
Mixed feeding	9 (24%)
Formula feeding	16 (43%)

Data are represented by the mean (standard deviation)//median (IQR range) or n (%). T1—time of first blood sample, near discharge. T2—time of second blood sample, on follow-up at 2–3 months’ corrected age. * 2 participants from one set of triplets, 3 from 3 sets of twins in which the other sibling did not participate, and 5 sets of twins. ** Calculated from the group of 32 infants who received PN. *** Includes one infant who switched to formula feeding from recruitment to T1. Among babies receiving mixed feeding, 2/5 had unfortified breastmilk and 3/5 were on fortifier.

**Table 2 nutrients-16-04220-t002:** Serum folate concentrations at time points T1 and T2 according to the type of feeding.

	**T1**		
	Mother’s Own Milk (MOM) N = 7	Mixed feeding/Formula N = 6	Fortified MOM N = 32
Mean (SD)	23.7 (9.8)	35.6 (8.1)	43.2 (3.3)
Median (range)	19.5 (14.7–41.2)	36.0 (22.0–44.4)	44.4 (31.7–45.5)
IQR	15.9–32.8	30.0–43.0	41.7–45.5
	**T2**		
	MOM N = 12	Mixed feeding N = 9	Formula N = 16
Mean (SD)	38.7 (5.1)	43.8 (1.2)	45.1 (1.0)
Median (range)	39.1 (30.1–45.5)	45.5 (42.4–45.5)	45.5 (41.9–45.5)
IQR	33.9–43.4	43.8–45.5	45.5–45.5

Data are serum folate concentrations, nmol/L.

## Data Availability

The datasets generated and/or analyzed during the current study are not publicly available due to privacy and ethical restrictions, but selected anonymized datasets may be available from the corresponding authors upon reasonable request.

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
