# Peer review of "Serum Folate Concentrations in Exclusively Breastfed Preterm Infants Who Received No Supplementary Oral Folic Acid After Discharge: A Prospective Cohort Study"

_nutrients, 2024, doi:10.3390/nu16234220_

Round 1
Reviewer 1 Report (Previous Reviewer 1)
Comments and Suggestions for Authors
The authors responded to my suggestions and the article has improved however some aspects remain critical:
1. Since there is no valid definition for folate deficiency or sufficiency in this age group, the title may read: Serum Folate Concentrations in Exclusively Breastfed Preterm Infants Who Received No Sup-plementary Oral Folic Acid After Discharge: A Prospective Cohort Study
2. The authors mentioned in the introduction and in their reply: “Preterm birth is associated with a risk of folate depletion due to increased demand from an accelerated growth rate and the need for rapid red blood cell production due to multifactorial anemia of prematurity.”. However, this statement is speculative and not supported by any evidence. All infants (term and preterm) have high demands for folate due to growth rate. That why the intake requirements for this age group is higher/per kg body weight compared to adults. I am not aware of literature showing that pre-term infants are at a higher risk of folate deficiency than term infants.
3. Physiological changes of folate after birth have been shown on a population level (example PMID: 14633879). Normally, women are recommended to use folic acid supplements in the first three months of pregnancy and in countries without folic acid fortification, the majority of women do not take folic acid in thirs trimester. The reason for physiological decline of serum folate after birth is not known, but could be related to expansion of blood volume and storing folate in tissues.
4. Again, the discussion about folate deficiency is not proven by measuring serum folate and comparing it with the WHO cutoffs that are indeed cutoff values for adults. Example: in the end of the introduction: “....... would develop subclinical folate deficiency....”. This is not rational; first because the infants were not clinically investigated to see if any symptom might be explained by folate deficiency, second this statement implies a range of folate for subclinical deficiency and another one for clinically relevant deficiency, and third because the definition of folate deficiency in this age group is not based on any objective evidence. Therefore, it is suggested to keep the study observational and descriptive and then compare with the literature to conclude on what could be the normal range of folate for this age group/health condition.
5. The authors may remove referring to “patients” and use instead “infants”; example: “Patients were recruited between January 2016 and April 2018.”
6. The point related to fasting was meant to be 1-2 hour after the meal, since folic acid peak is reached before that time. This was not planed/done in the study and was not justified in the previous version. Possibly because the authors did not make thoughts about this point or due to simplification reasons. However, it is not possible to retrospectively argue whether this was due to ethical issues. This argument should have been done in the planning phase. The statement should be deleted “The time of sampling was not standardised .......... due to ethical concerns around fasting ex-preterm infants.”
1. The number of multiple births in this study on preterm infants is exceptionally high. In addition, single infants from multiple births were recruited. This could imply issues related to selection of infants into the study (why one infant was recruited and the other one not recruited) and independency of the observations (several dependent observations). The authors provided some arguments in their reply to my previous comment. However, the text is not reporting any argument that justifies recruiting twins in this study.
Author Response
Thank you for your comment. Please see attached document.

Reviewer 2 Report (Previous Reviewer 2)
Comments and Suggestions for Authors
We have already checked this when you submitted it last time, so there is nothing in particular that we need to point out again this time. The limitation has also been discussed, so there is nothing in particular that we need to request you to revise.
Author Response
Thank you for your comments and for taking the time to review our manuscript.
This manuscript is a resubmission of an earlier submission. The following is a list of the peer review reports and author responses from that submission.
Round 1
Reviewer 1 Report
Comments and Suggestions for Authors
The study presents baseline and longitudinal data on serum folate concentrations from a cohort of preterm infants who received different sources of folate. The study is addressing an important topic and the data are novel and of high scientific value.
The hypothesis, data analysis and presentation are insufficient. The conclusion is too strong, given that it is based on an invalid assumption, that the WHO population reference ranges of serum folate concentrations can be extrapolated to preterm births. The concentrations of serum folate in pre-term infants from the present study are comparable with those from pre-term and term infants from other European populations not exposed to folic acid supplementation or fortification. The conclusion is not justified by the data and may be easily misinterpreted. I suggest presenting the results in a descriptive way, without speculations on deficiency or over-supplementation of folate.
Specific comments
1. The information on ethical approval and consent are usually not reported in the abstract.
2. Please use SI units to present folate concentrations all over the manuscript.
3. Page 2, line 86, Reference 8 is a review article and can not be cited here for a key statement. Please use original studies on breastmilk folate that were used to define the AI for formula milk.
4. Mean folate concentrations in Norwegian infants was between 47-55 nmol/L, thus even higher than in the present study with no differences between breastfed and non-breastfed infants (Am J Clin Nutr. 2008;88:105–14). It is not clear why to think that pre-term infants may be depleted after release from the NICU unit, especially if they are not exclusively breastfed?
5. Serum folate concentrations are high at birth and show a physiological decline afterward up to 24 month of age. It is not known whether these longitudinal changes follow the same pattern in preterm and term infants.
6. Serum folate is influenced by recent folate intake and folate levels peak 1-2 hours after the meal. This makes interpretation of serum folate collected in preterm babies with frequent feeding sessions very difficult. Was the time interval between the meal and the blood sampling standardized in this study?
7. Serum folate reflects recent folate intake. Especially in small babies with frequent feeding and different physiology of gastric motility, the absorption of folic acid can be very fast and can cause peaks/fluctuations in serum folate. Thus, serum folate in babies is even less optimally suited to reflect folate status (vs. adults). Folic acid can cause stronger fluctuations in serum folate than folate from human milk (that is bound to proteins). This factors need to be considered in the interpretation of the results.
8. Page 2, Lines 81-82: it would be good to elaborate on the arguments behind this wide range of intake 23-100 µg/kg/d recommended by ESPGHAN.
9. One factor that needs to be taken into consideration is that human milk may contain less folate than formula milk, but it also contain high levels of folate binding proteins that may facilitate folate absorption from human milk (compared to formula milk).
10. Page 3, materials and methods: should specify the months/years of recruitment (not in the results section).
11. Due to complexity and heterogeneity of the diet during the follow up, a study flow diagram or a study plan showing the dietary groups and sample size would be helpful as a supplemental data file.
12. The participants should be better characterized. Table 1 is not informative. The population of infants shown in Figure 1 is expected to differ by several characteristics (gestational age, %IUGR, weight, sex, complications during pregnancy). The baseline characteristics should be additionally presented by the feeding groups.
13. What are the risk factors that likely related to preterm (such as maternal smoking, supplement use during pregnancy, obesity, pre-eclampsia, etc)? What about the ethnic origin of the infants, given the population diversity in the UK? Potential differences in characteristics should be formally tested. The characteristics of the population during T2 visit are equally important to see if the infants from different nutrition groups had similar growth rate, …etc.
14. Observations from multiple (twin or triplet) (33% of the cases) cases are not independent and cannot participate multiple times in the sample size. The multiple pregnancies were exposed to the same folate level in utero (and possibly postnatal?) and their folate status is likely correlated. This has implications for the statistical tests applied.
15. Please add the average of added folic acid as µg/kg/d to the descriptive tables in each dietary group.
16. As shown in Figure 1, the serum folate concentrations were extremely heterogeneous at T1 (5-20 ng/ml, large variance). Whereas, the variability of the concentrations at T2 was obviously less (12.5 -20 ng/ml), although the infants have received different nutrition after birth and after discharge. This suggests that the reasons for heterogeneity lay most likely before or at birth.
17. It is clear that the study is not interventional, but it is important to elaborate on possible factors that influenced the type of the diet during the stay and after the release from the NICU. Why some babies were taking formula and others received fortified MOM (is this human milk from the mother of the same infant or from human milk donor?)? It is very likely that the population characteristics are different.
18. In figure 1, there is a segregation in the data at T1. There seem to be some babies (n = 6?) with very low folate at T1 compared to the rest of the group. At least 5 of those experienced a dramatic raise in serum folate at T2. What was the diet group in those infants?
19. Page 3, lines 139-142: the WHO reference ranges are for the general population. There are no established reference intervals for newborns (term or pre-term). However, it is well known that newborn infants have higher folate than their mothers and than adults population (Example; PMID: 39326699). The average serum folate in term babies from Spain or other European populations is 15-17 ng/ml or 35-40 nmol/L in a population without folic acid fortification or high dose supplemental folate. These concentrations are comparable to those observed in the present study.
20. The results of serum folate at T1 and T2 are not shown for the same infants (longitudinal changes according to the diet). This makes it difficult to observe follow up changes for infants on the same diet. The longitudinal changes and comparisons are needed. The longitudinal changes need to be adjusted for baseline serum folate concentrations and type of the diet.
21. The statistical analysis mentioned repeated measure tests; which statistical tests was applied? I cannot see which results are related to this test. the samples size should be also clear. Maybe better to run a regression analyses on the change of serum folate and enter the diet model, together with other adjustment variables to the model. An interaction between the baseline serum folate and the diet would be informative.
22. Page 5, line 190-196: the coefficient of associations and R2 statistics are shown without any explanation. What do the results mean?
23. I assume (especially due to assigning a high level for some samples) that the continuous variables are not normally distributed and t-test or ANOVA test cannot be applied without log-transformation or should be replaced by non-parametric tests.
24. This reviewer think that multivariate regression analysis cannot be run on multiple pregnancies that are not independent observations.
25. Figure 1 is (according to the aim of the study) the most important part of the results but the less informative part. The data presentation is not sufficient. Figures can be added as online supplements, but researchers look usually for mean (SD) or other measures that can be combined in for instance meta-analysis or be used to compare with previous studies. It makes no sense to use the WHO Cutoffs. This will cause confusion and raise the wrong conclusion that infants are receiving too much folic acid. But I can not see any evidence that the infants have received too much folate or the current guidelines are overestimating the folate requirements.
26. How were the hematological index (eg. hemoglobin), infant weight, height, and head circumference z scores developed between T1 and T2 according to the diet or according to the change in serum folate?
27. I expect the authors to compare their results with previous studies on preterm births (Vanier TM and Tyas JF, 1966; Hibbard ED, 1974; Roberts PM, 1969; Roberts PM, 1972; Ek J, Behncke L et al., 1984).
Reviewer 2 Report
Comments and Suggestions for Authors
Neither at the time of discharge from the NICU nor at the time of follow-up (2-3 months later) were there any cases of folate deficiency, and in newborns who received folate-containing nutritional products, an increase in serum folate levels was frequently observed. This fact is valuable.
We would be grateful if you could answer the following questions.
Was the folate content of the breast milk of the participants in this study measured?
Was the nutritional status of the breastfeeding mothers who gave breast milk recorded (regarding folate intake)?
Please discuss whether the specificity of the area surveyed is related.
Was there any intake of foods high in folate (vegetables, liver)?